# Melting of subducted sediments reconciles geophysical images of subduction zones

M. W. Förster [ID] [1✉] & K. Selway[1]

Sediments play a key role in subduction. They help control the chemistry of arc volcanoes and the location of seismic hazards. Here, we present a new model describing the fate of subducted sediments that explains magnetotelluric models of subduction zones, which commonly show an enigmatic conductive anomaly at the trenchward side of volcanic arcs. In many subduction zones, sediments will melt trenchward of the source region for arc melts. High-pressure experiments show that these sediment melts will react with the overlying mantle wedge to produce electrically conductive phlogopite pyroxenites. Modelling of the Cascadia and Kyushu subduction zones shows that the products of sediment melting closely reproduce the magnetotelluric observations. Melting of subducted sediments can also explain K-rich volcanic rocks that are produced when the phlogopite pyroxenites melt during slab roll-back events. This process may also help constrain models for subduction zone seismicity. Since melts and phlogopite both have low frictional strength, damaging thrust earthquakes are unlikely to occur in the vicinity of the melting sediments, while increased fluid pressures may promote the occurrence of small magnitude earthquakes and episodic tremor and slip.

[1] Australian Research Council Centre of Excellence for Core to Crust Fluid Systems (CCFS) and Department of Earth and Environmental Sciences, Macquarie University, New South Wales, Australia. ✉email: michael.forster@mq.edu.au

Subduction is a fundamental process controlling the chemical evolution of our planet. At most subduction zones, a package of sedimentary rocks survives being scraped off at the trench and is subducted[1]. Although they are a volumetrically minor component of the subducting slab, these sediments are compositionally distinct from the igneous and metamorphic rocks beneath them and provide unique input to the subduction system[2]. The sediments are compacted and start to de-water at shallow depths (<10 km) and hydrous serpentinite minerals form in the overlying mantle at temperatures less than ~700 °C[3,4]. These processes have been investigated in detail, partly because the fluid release and serpentinite formation may control the depth range of damaging thrust earthquakes in subduction zones and explain the seismic properties of the mantle wedge[5]. The remaining portion of the subducted sediments will melt, often making a chemically important contribution to arc melts[2]. As discussed below, in many subduction zones this melting will occur at depths between fore-arc serpentinites and the source region of arc magmas.

The precise conditions for melting of the subducted sedimentary material will depend on pressure and the chemical composition of the sediment but melting may begin at temperatures as low as 675 °C[6] (Fig. 1a). Although sediments dehydrate early during subduction, a continued flux of fluids expelled from underlying serpentinites promotes sediment melting[7]. Partial melts of the subducted sediments eventually separate from their source, rise, react with and hybridise the mantle wedge beneath arcs to generate subduction-related magmatism[8]. The range of calculated[9] subduction zone slab top temperatures show that in many subduction zones, hydrous sediments will begin to melt in the fore-arc region (Supplementary Fig. 1). Some subduction zone thermal models indicated by exhumed blueschists and eclogites[10] are 200–400 °C higher than these calculated models, which would

extend the area of sediment melting even further into the fore-arc. When slab top temperatures are such that subducted sediments begin to melt in the fore-arc, the resultant volatile-bearing silicic melt (sediment melt) will infiltrate and metasomatise the overlying mantle. Importantly, this can occur where mantle temperatures are well below <1000 °C, and thus, still too low to generate the more mafic basaltic to andesitic melts that are characteristic of arcs (arc melt).

In this study, we propose that a phlogopite-pyroxenite metasome is responsible for the highly conductive magnetotelluric anomaly imaged within the fore-arc of many subduction zones (Figs. 2–4). As subduction proceeds and more sediment is continually provided to the system the metasome will grow, such that at any snapshot in time the existing metasome will be likely to contain a small amount of sediment melt. At shallower depths in the fore-arc, temperatures fall below the solidus of sediment melt (less than ~675 °C) and crystallisation leads to the exsolution of a saline fluid. The depth and extent of the metasome will depend on sediment composition and subduction zone geotherm and geometry. For instance, where sediments are more hydrous or the subduction zone geotherm is hot, there may be more lateral distance between sediment melting and arc melting. However, in steeply-dipping subduction zones the sediment melting and arc melting regions may essentially overlie one another. In such a situation the metasome is unlikely to develop at depth but arc magmas may be more K-rich as they have a higher component of sediment melt which would have been otherwise trapped within the fore-arc metasome.

## Results and Discussion

**Reaction experiments.** To consider the composition of the metasome produced by the interaction of the sediment melts with

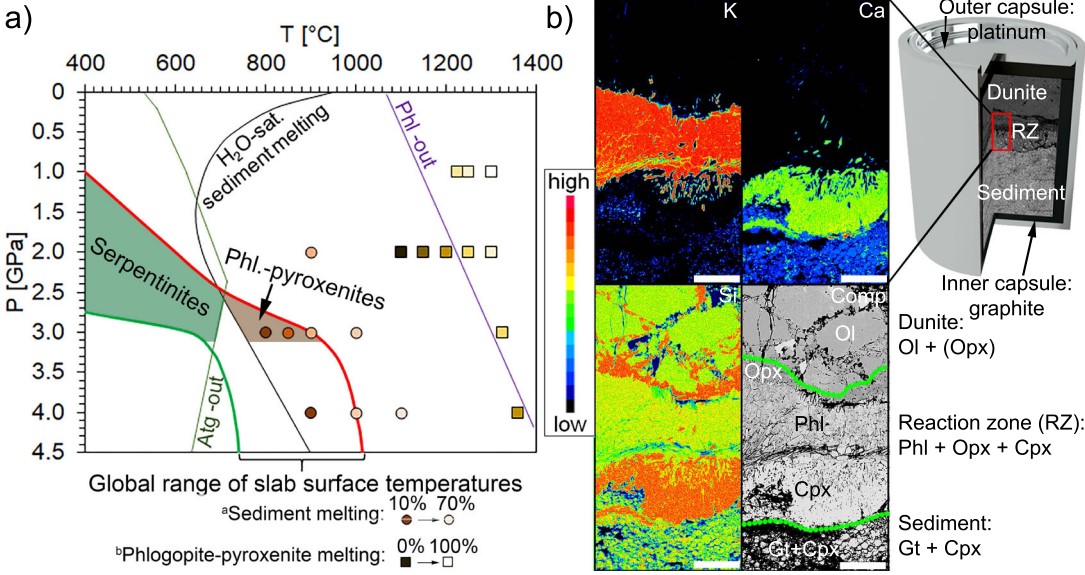

**Fig. 1 Sediment melting experiments compared to global slab surface temperatures (a) and element maps of sediment-peridotite reaction zone products (b). a** The global range of slab surface temperatures (area between green and red lines, D80 model of Syracuse et al.[9]) overlaps with the $H_2O$ saturated sediment melting curve[36] within the fore-arc mantle, in front of the slab-mantle coupling region. Subducted sediments cross the solidus of $H_2O$-saturated sediment melting at ~2.5–3.2 GPa and ~675–950 °C to react with mantle peridotites, producing phlogopite-pyroxenites (grey shaded area). Reaction experiments confirm sediment melting at fore-arc mantle conditions (circles)[11,12] and the stability of phlogopite-pyroxenites at T < ~1100 °C (squares)[45,46]. At lower pressure and temperatures, serpentinites are present within the stability field of antigorite (green shaded area). Both the serpentinite and phlogopite-pyroxenite fields are confined to >3.2 GPa, which translates to ~100 km and should be regarded as the maximum thickness of the fore-arc lithosphere. **b** Reaction experiments demonstrate that the interaction of sediment melt with depleted peridotite (dunite) produces phlogopite-pyroxenites and enriches the former peridotite in K, Ca and Si. The location of the reaction zone is given on a rendered image of the capsule, which is about 4 mm in diameter. Scale bar corresponds to 100 μm. Atg Antigorite, Phl Phlogopite.

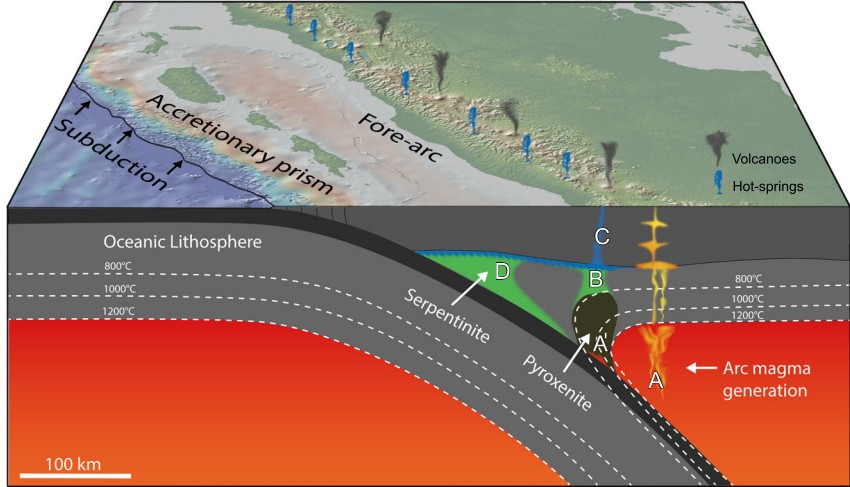

**Fig. 2 Subduction zone with fore-arc metasomatism model.** Arc magmas form below the volcanic front (**A**), where fluid-fluxed melting of mantle peridotites occurs. In front of the region of arc magma generation (A') subducted sediments melt and their hydrous melts rise and react with peridotite. Since temperatures in front of the arc (A') are below the solidus of peridotite, hydrous sediment melt reacts to form a phlogopite-pyroxenite metasome. Fluids expelled by phlogopite-pyroxenite formation rise through the mantle wedge **B** into the crust **C** to form reservoirs and hot springs. Map modified from GeoMapApp, www.geomapapp.org/ with Global Multi-Resolution Topography (GMRT)[47].

the overlying mantle, the present study focusses primarily on our geophysical model, using well pre-characterised reaction experiments[11,12] with readily available phase compositions and microstructures. In this experimental setup, sediment subduction is simulated by placing a layer of depleted mantle peridotite directly on top of marine sediment in a 50/50 ratio in the same capsule. Pressure (3 GPa) and temperatures (800–900 °C) resemble conditions of the subduction zone fore-arc adjacent to the volcanic arc (Fig. 1a). The experimental product exhibits a reaction zone of phlogopite-pyroxenite metasome between the sediment and depleted peridotite with a thickness of ca. 500 μm (Fig. 1b), which increases with temperature, melt fraction and duration of an experiment[11,12]. A representative reaction experiment is depicted by means of EPMA maps in Fig. 1b. The reaction concentrates all potassium within a layer of phlogopite (K map in Fig. 1b), although all potassium is contained in the sediment before the start of the experiment. The reaction zone is also enriched in Ca and Si, present within clinopyroxene and orthopyroxene. Since sediment melt accommodates more volatile components (5–10%) than phlogopite-pyroxenite (<5%), the reaction will also produce a fluid phase. The high-pressure experiments do not account for fluid loss before sediment melting and volatile contents could be overestimated. Nevertheless, in subduction zones, the volatile content of this sediment melt should be above >5% due to the continuous devolatilization of serpentinites[4,13] that underlie the sediments, which are an additional fluid source that is not accounted for in the reaction experiments. Since fluids have high solubility in silicic melt[14] and partial melting of sediments produces <10% melt[11], even low amounts of fluid (~0.5–1%) from the underlying slab translate to 5–10% volatiles within the sediment melt.

**Melting sediments explain geophysical models of subduction zones.** Magnetotellurics (MT) is a geophysical technique ideally suited for investigating subduction zones since it is sensitive to conductive phases such as melts and fluids. However, MT models over subduction zones show that the strongest conductive feature is generally not beneath the volcanic arc but is instead offset in the fore-arc[15,16]. Subduction zone conductors have been interpreted in several ways, including an accumulation of dehydration

fluids[15], release of fluids at the basalt-eclogite transition[16] and a non-linear pathway for arc melts through the mantle wedge[17].

We propose that melting of sediments in the fore-arc provides a consistent, genetic explanation for the strong fore-arc conductor in MT models. The phlogopite-rich metasome is expected to have a particularly strong conductive response because phlogopite has been experimentally shown to be anomalously conductive[18,19]. Indeed, the reaction experiments produce phlogopite with high fluorine contents, which has an especially high conductivity[17]. We produced synthetic models of expected electrical resistivity structures for the Cascadia (western North America) and Kyushu (southern Japan) subduction zones to compare with MT inversions, using existing geotherm models and experimental conductivity studies (see Methods). These two regions were chosen because they have well-defined MT characteristics and geotherms and they also illustrate the behaviour of subduction zones with different geometries.

Theoretical resistivity structures were calculated based upon simplified subduction zone compositions of mantle peridotite, phlogopite-pyroxenite metasome, and 1% sediment melt, 1% arc melt, and 1% saline fluid in the appropriate regions (see Methods, Supplementary Table 1 and Figs. 2–4). In the temperature range of interest, the phlogopite-pyroxenite is more conductive than peridotite and the saline fluid is more conductive than arc melt. The MT responses of the resistivity structures were forward modelled and these MT responses were then inverted. This procedure was chosen because MT inversions are non-unique and it allowed us to directly compare our inversion of synthetic data with the inversion of field data. These models are simplified and do not seek to model the full complexities of potential subduction zone conductive anomalies but are specifically designed to model the anomalies arising from the melting of subducted sediments.

Synthetic modelling of the Cascadia subduction zone was carried out along the CAFE line and results are compared with the MT model of McGary et al.[17] (Fig. 3). A sharp boundary was allowed along the slab surface in the inverse synthetic model, following previous modelling. The main features of the models compare extremely well, with a strong conductor overlying the slab and rising towards the surface in the fore-arc, ~20 to 30 km trenchward of the arc volcano (Mt. Rainier). The most conductive

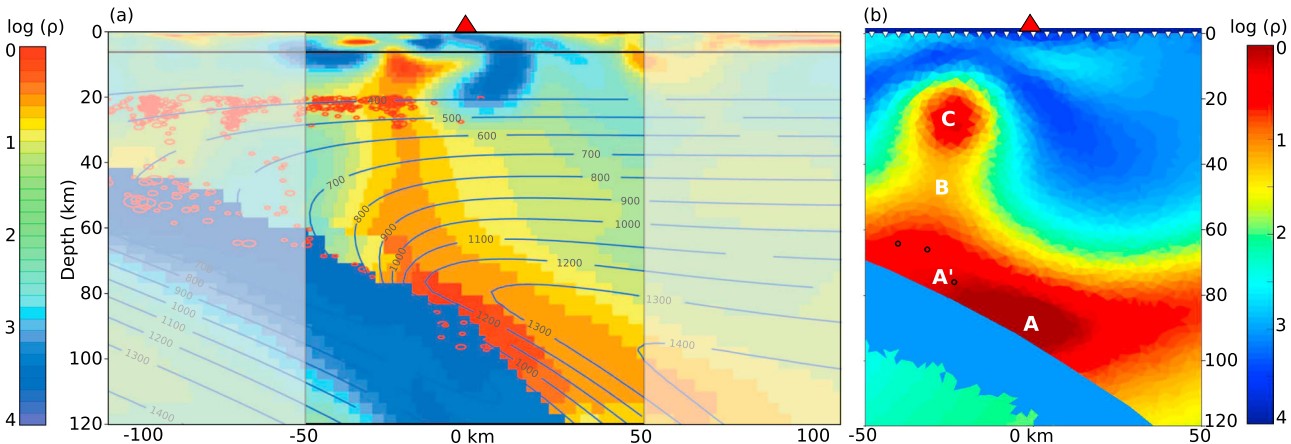

**Fig. 3 Original (a) and synthetic (b) MT models for the Cascadia subduction zone.** Colour scale is log electrical resistivity ($\rho$). **a** Shows the inverse MT model along the 'CAFE' line in the Cascadia subduction zone, western North America, adapted from McGary et al[17] with the region of interest highlighted. Reprinted by permission from Springer Nature, Nature, Pathway from subducting slab to surface for melt and fluids beneath Mount Rainier, McGary et al[17], ©2014. **b** is the synthetic MT model, which was created by forward modelling and then inverting the MT responses from a subsurface with conductivities defined by mantle peridotite, arc melts, and the phlogopite-pyroxenite metasome, melt, and saline fluids produced by melting of subducted sediments. The synthetic model reproduces the main conductive features, which are offset trenchward of Mt. Rainier (red triangle). Labels (A, A', B and C) follow the regions labelled in Fig. 2. Black circles in (**b**) show the locations of small (magnitude < 3) earthquakes that have occurred in the mantle wedge since 1980.

regions are modelled just above the slab surface (regions A and A'), induced by high temperatures and sediment melt, and from ~10–30 km depth 20–30 km trenchward of the volcano (region C), due to the presence of saline fluids produced in the crystallisation reaction of the phlogopite-pyroxenite metasome. Although arc melt is included in a column extending up to the arc volcano, this region is not associated with a strongly conductive anomaly. This is primarily because the conductivity of arc melt is significantly lower than saline fluid at crustal and uppermost mantle temperatures (Supplementary Fig. 2). Seismic models of the CAFE line region[20] are consistent with this interpretation, with slow seismic velocities modelled both directly beneath and ~15–20 km trenchward of Mt. Rainier. These slow velocities align with our proposed regions of arc melt and saline fluids, although the interpretation of the seismic model was that they are both caused by melt. Since saline fluids are more conductive than arc melts at crustal temperatures, the conductive response of the saline fluids dominate the MT model. Non-uniqueness in the MT inversion process has converted the broad, moderately low resistivity zone caused by arc melts in the forward model (Supplementary Fig. 3a) to a thinner, lower resistivity zone at greater depth in the inverse model (Region A in Fig. 3), while maintaining a good fit to the data.

Synthetic modelling of the Kyushu subduction zone was carried out along a profile in southern Kyushu (Fig. 4), approximately coincident with the profile crossing the Kirishima arc of Hata et al[21]. Following the original inversions, the slab surface was not included as a sharp boundary. The models agree well in the upper ~50 km (Fig. 4), with a shallow conductor rising from the slab slightly offset trenchwards from the arc volcano. The Kyushu subduction zone dips more steeply than Cascadia so there is less lateral distance separating the regions of sediment melting and the source region of arc magmas.

At depths greater than ~50 km the agreement between the synthetic and original Kyushu models is poorer and the synthetic model shows a strong conductor from ~60 km depth that is not observed in the original model. This discrepancy appears to be mainly due to a lack of model resolution at this depth. The original Kyushu model (Fig. 4a) is derived from widely-spaced network-MT stations that extend to periods of >20,000 s[22] together with geodynamic depth sounding data that are denser

but have a more limited period range (up to 1000 s). Resolution of features at depths >50 km is largely from the network-MT data and tests run by Hata et al.[22] show that resistivity is poorly resolved at these depths, with resistivities up to an order of magnitude higher and lower than those in the preferred model producing acceptable data fits. The synthetic model contains a broader period range ($10–10^5$ s) at more densely spaced stations (4 km) but still shows poor resolution in this part of the model, highlighted by the difference between the forward (Supplementary Fig. 3) and inverse (Fig. 3) models. The original and synthetic models were run with different inversion schemes that appear to have responded to this lack of resolution by imposing more resistive and more conductive features, respectively. Therefore, while we consider that while the shallower features in Kyushu support our proposed model, the deeper features do not provide so much information. In addition to aiding interpretations of MT models, these results also clarify some of the differences between seismic and MT models of subduction zones. For instance, along the Cascadia CAFE line, the lowest seismic velocities are associated with the cold subducting slab and the serpentinized mantle wedge to ~45 km depth[23]. Serpentinite minerals are not strongly conductive, so there is little MT response at shallow depths. At greater depths, the highly conductive F-rich phlogopite does not produce a corresponding seismic velocity anomaly so the metasome is not clearly observed in the seismic models. The small proportions of melt and fluid that are likely to be present will not produce a very strong seismic or electrical response, with the exception of the saline fluid produced during metasome formation, which is highly conductive even at low temperatures[24].

**Implications for K-rich volcanism and seismicity.** The shallower depth of sediment melting than arc melting in many subduction zones has several significant implications. Geochemically, the phlogopite-pyroxenite metasome, produced when sediment melts metasomatize the overlying mantle, will be potassium-enriched[11,25,26]. This metasome is a likely source for lavas of exotic potassic to ultrapotassic composition which are strongly enriched in large ion lithophile elements, high field strength elements and rare earth elements. In addition to potassic magmas from post-collisional settings[27], the formation of phlogopite-pyroxenites within the fore-arc mantle is reported from Mexico[26]

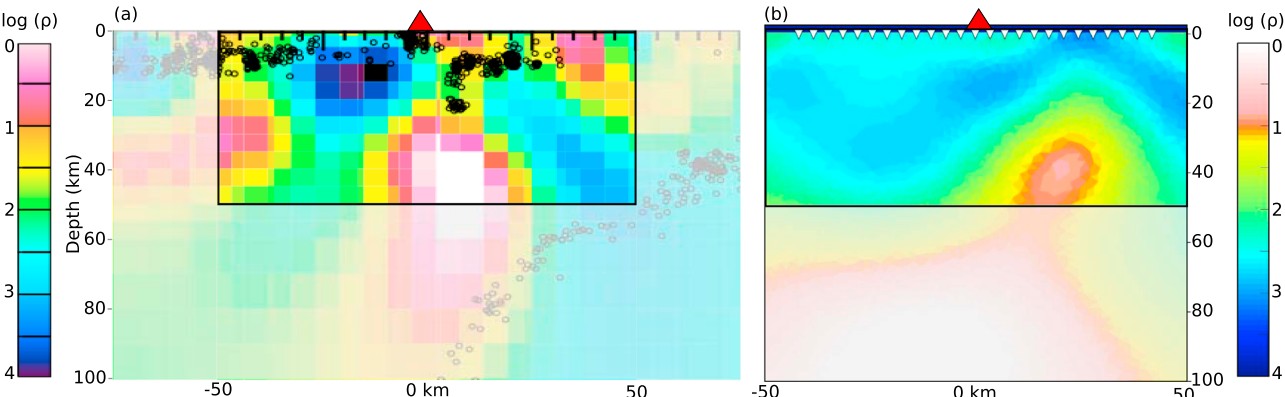

**Fig. 4 Original (a) and synthetic (b) MT models for southern Kyushu.** Colour scale is log electrical resistivity ($\rho$). **a** Shows the inverse MT model in southern Kyushu crossing the Kirishima arc volcano from adapted Hata et al[21] with the region of interest highlighted. Reprinted by permission from John Wiley and Sons, Journal of Geophysical Research: Solid Earth, 3-D electrical resistivity structure based on geomagnetic transfer functions exploring the features of arc magmatism beneath Kyushu, Southwest Japan Arc, Hata et al[21], ©2017. **b** is the synthetic model, produced from a subsurface with conductivities defined by mantle peridotite, arc melts, and the phlogopite-pyroxenite metasome, melt, and saline fluids produced by melting of subducted sediments. The synthetic model approximately reproduces the conductor extending to ~50 km depth just trenchward of Kirishima volcano (red triangle). Shading at depths >50 km reflects poorer model resolution at these depths.

and is likely a common phenomenon in subduction zones, though only visible when magmatic activity shifts from the arc into the fore-arc and potassic magmas are erupted[25]. In Kyushu, subduction activity has shifted into the fore-arc due to slab rollback and steepening of the subduction zone[28]. MT models depict the high conductivity anomaly slightly in front of Kirishima volcano in southern Kyushu (Fig. 3), while MT models from Aso volcano in northern Kyushu do not image a high conductivity anomaly[21,22]. Intriguingly, Aso volcano erupts lavas that follow a high-K calc-alkaline to shoshonitic trend[29], while Kirishima extrudes lavas of common calc-alkaline composition[30]. According to the model presented in our study, Aso probably sources its magma from the metasomatized fore-arc mantle which is nearly exhausted and thus not visible on the MT image, while Kirishima has not yet started to erupt lavas from the metasomatized fore-arc. Slab rollback also currently occurs at the intersection of the Izu–Bonin and Mariana arcs where the 'alkalic volcano province' produces high-K calc-alkaline and shoshonitic composition[31]. In contrast, both the Izu–Bonin arc north of the 'alkalic volcano province' as well as the Mariana arc to the south produce low-K calc-alkaline magmas. This region has not yet been imaged with MT but this would be an important test of our model. Another useful test would be the MT response of sediment-starved subduction systems, which would be predicted to show little to no conductive metasome.

Geophysically, as well as providing an explanation for MT models, sediment melting should influence the location and depth range of subduction zone earthquakes. Thrust earthquakes, which are the main seismic hazard in subduction zones, generally do not extend into the serpentinite field because serpentinite minerals have weak stable-sliding behaviour[5]. Phlogopite is also likely to have low frictional strength[32] which, together with the presence of melt and fluid, would be expected to reduce the likelihood of destructive thrust earthquakes in the metasomatised fore-arc mantle wedge. In contrast, small magnitude intra-slab and mantle wedge earthquakes could be generated by the increased fluid pressure caused by fluids and melt released by the melting sediment[33]. Several small (magnitude < 3) earthquakes have occurred where sediment melt ascends into the Cascadia mantle wedge (Fig. 3). Fluids and melt could also promote the occurrence of episodic tremor and slip and clear correspondence between subduction zone mantle wedge conductors and regions of concentrated tremors have been observed in northern California[34]. A full understanding of subduction zone seismicity requires accurate models of fluid sources and movements[35] and the recognition of sediment melts is an important contribution to this picture.

## Methods

**Experiment starting materials.** Sediment/peridotite reaction experiments and their implications on major and trace element fractionation have been discussed in detail previously[11]. The experimental setup consists of two starting materials, a sediment and a depleted peridotite (dunite) in a two-layer arrangement with the depleted peridotite placed on top of the sediment. The sedimentary starting material is a siliciclastic marine sediment which was acquired from the International Ocean Discovery Project (IODP) site ODP 161–976 B 18 ×3 105–106.5. It contains <10% carbonate and 5–10 wt% $H_2O$ as estimated by difference to 100 wt% in glasses measured by using an electron probe microanalyzer (EPMA). The dunite is a sample (ZD11–53) from the Zedang ophiolite (south Tibet, China) and contains olivine (>97%), spinel (~2%) and clinopyroxene (<1%). Both samples were powdered in an agate mortar.

**Experimental and analytical techniques.** The experiments were performed using a piston cylinder apparatus at the University of Mainz. Thermobaric conditions applied were 2–3 GPa and 750–900 °C, which correspond to the fore-arc setting of a subduction zone. After each experimental run, the capsule was longitudinally cut in half, embedded in epoxy, and polished for characterisation of the charges. Major element contents of experimental run products were acquired using a JEOL JXA 8200 Superprobe EPMA equipped with 5 wavelength dispersive spectrometers at the University of Mainz, Germany.

**Magnetotelluric modelling.** Subduction zone geotherms and slab top temperatures were taken from existing models[9,23,36], with the specific CAFE line Cascadia thermal structure taken from van Keken et al.[37]. The subduction zone conductivities were forward modelled with simplified compositions comprising anhydrous harzburgite (20% pyroxene and 80% olivine) apart from the phlogopite-pyroxenite metasome (20% phlogopite and 80% pyroxene). We assume that melting of subducted sediments begins where the slab top temperature reaches 675 °C. The mantle overlying this region is assumed to be composed of metasome and to contain water-rich sediment melt (12% water) until the temperature decreases below 700 °C. At this temperature, all melt will have crystallised to metasome plus saline fluid. Therefore, at shallower depths, the mantle wedge is composed of harzburgite with a small proportion of 5% NaCl fluid[24]. Arc melting was modelled to begin at a temperature of 800 °C, an average value within the estimated range of ~725 to 900 °C[38] and to extend for a lateral distance of ~50 km. Arc melts, also containing 12% water, were assumed to exist through the entire overlying section of mantle and crust, which was modelled with a homogenous 20% pyroxene and 80% olivine composition. Throughout the model region, melt and fluid phases extended to a depth of 10 km, assumed to be the brittle–ductile transition, though this may vary by location and upper crustal structures may allow for additional upward and lateral fluid flow that would change the shallow resistivity structures. A 500 m thick, 500 Ωm layer was inserted at the surface to mimic a sedimentary basin.

Electrical conductivities of the different components were calculated from experimental results as follows: olivine from Gardes et al.[39], pyroxene from Dai and Karato[40], sediment melt and arc melt (containing 12% water) from Sifre et al.[41], phlogopite from Li et al.[19] and saline fluid (containing 5% NaCl) from Sinmyo and Keppler[24,42]. One percent melt and fluid was used in most calculations but tests were also run for the Cascadia model with 0.5% and 5% melt (Supplementary Fig. 3) with similar results. A modified Archie's Law was used to geometrically combine the components[40], assuming good connectivity of the conductive phase as expected for 1% melt or fluid (Laumonier et al[43]; Supplementary Fig. 5).

Once the predicted resistivity structures of the subduction zones were compiled (Supplementary Fig. 2), MARE2DEM[44] was used to forward model the expected MT responses at 25 synthetic stations spaced at 4 km intervals (white triangles in Figs. 3 and 4) over 17 periods ranging from 10 to $10^5$ s. Small (5%) error bars were added to these forward modelled responses and MARE2DEM was used to invert them to allow a direct comparison to be made with the published inverse models. Inversions ran from a homogenous, 100 Ωm half-space starting model. The Cascadia model (Fig. 3) inverted to a root mean squared (rms) error of 1.5 and the Kyushu model (Fig. 4) inverted to a rms error of 2.3. Both colour scales were set to most closely match the previously published models.

## Data availability
The authors declare that the source data underlying the main figures (Figs. 1, 3, and 4) of this study are available from the corresponding author upon reasonable request.

## Code availability
The authors declare that the code underlying Figs. 3 and 4 of this study is available from the corresponding author upon reasonable request.

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

## Acknowledgements

We are grateful to Peter van Keken and Geoff Abers, who generously provided the geothermal models, and to Stephen Foley and Helen Janiszewski for proofreading the manuscript. We thank Qing Xiong for providing the dunite sample ZD11–53 used in the experiments. The International Ocean Discovery Project (IODP) is acknowledged for providing the Mediterranean marine sediment sample from site 161–976. This is contribution 1550 from the ARC Centre of Excellence for Core to Crust Fluid Systems (www.ccfs.mq.edu.au) and 1415 in the GEMOC Key Centre (www.gemoc.mq.edu.au). MWF was supported by ARC grant FL180100134. K.S. was supported during this work by ARC grant FT150100541. We thank three anonymous reviewers for their detailed comments that helped to improve this paper.

## Author contributions

M.W.F. and K.S. designed the study and wrote the manuscript. Both authors carefully edited the final version of the manuscript.

## Competing interests

The authors declare no competing interests.
