## [Peer Review File · Nature Communications]

REVIEWER COMMENTS

Reviewer #1 (Remarks to the Author):

The paper 'Melting of subducted sediments reconciles geophysical image of subduction zones' presents a study which aims to explain a globally observed conductivity anomaly in the trench ward side of the volcanic arc. The authors explain the conductivity anomaly to be caused by exsolution of saline fluids during crystallization of a phlogopite-pyroxenite metasome, which in turn is related to sediment melting. The authors present a synthetic MT modelling study, in which they generate a conductivity model based on petrophysical assumptions including the phlogopite-pyroxenites metasome and temperature models of the Cascadia and Kyushu subduction zone. For these conductivity models, MT forward responses are calculated and then inverted. The resulting conductivity model is then compared to conductivity models derived from real data for these two subduction zones. Agreement of the conductivity anomaly observed in the real and synthetic inversion model is then interpreted as evidence that the conductivity anomaly is related to sediment melting. Sediment melting is also proposed as a possibility to explanation for the existence of K-rich volcanic rocks in conjunction with slab-rollback events.

The paper is certainly of sufficient interest and novel to be published by a high impact journal like Nature Geoscience since it explains a globally observed anomaly of enigmatic origin and contributes to further understand, define and quantify hydration and dehydration processes of subduction zone. It is also clearly structured and well written and documented, in fact I enjoyed reading the paper. I therefore recommend it for publication after some amendments or discussions listed below are included in the document.

1) I think the paper would benefit from reasoning the strategy that the author's follow in presenting their 'evidence'. The authors go from a temperature and compositional model (geological model) to a resistivity model (physical parameter model), then translate that into data (data space) and reinvert the data to a physical parameter model and compare this with the derived observed physical parameter model. I do appreciate that the authors take into consideration that the shape of the physical parameter model very much depends on the data and inversion algorithm as it gives us only a biased or 'smeared' look at the Earth and it is that image which we should compare to the inversion model of the data. However, I think the paper would benefit if you would explain your overall strategy and most importantly reason why you do this. I think this explanation is necessary since you deviate from the trodden path in which one takes the physical parameter inversion model and interprets the anomalies with regard to temperature and/or composition and/or fluid content.

2) Please discuss the uncertainties in resistivity model that is generated (supplementary section). You make various assumptions about the composition, sediment volume and temperature (the effect of melt volume has been investigated). A large uncertainty will arise from the effective medium theory chosen to calculate the overall bulk conductivity of the mix. I would like to see some more detailed explanation/reasoning, how the resistivity model may vary depending on different assumptions/choices made and why the authors have chosen the particular values of the parameters. Also, do you see any possibility to derive a exsolved water volume from the observations directly which could be applied to MT studies on other subduction zones, i.e. convert the physical parameter to relevant geological parameters and if so, with which trade offs? This would be particularly interesting as it would help to interpret MT resistivity models at other subduction zones.

3) I am somewhat puzzled by the inversion results of the synthetic Cascadia data. There is a strong anomaly in the predicted conductivity structure associated with the phlogopite/pyroxenite sediment melt. However, in the inversion model the high conductivity anomaly is not visible at the same

location anymore but seems to have moved towards profile kilometer 10 or 20 and to a depth of 90 km. This should be explained, particularly since I do not see the same effect for the Kyushu case. Also, it would be great if you could include a bit more detail about the number of Mt stations you simulate as well as the frequency range in the data. What was the starting model that you chose?

4) In the paper of Worzewski et al., the anomaly was always found at a consistent depth and distance from the volcanic arc, independent of the convergence rate, subducting plate age or style (erosional/accretionary) of the subduction. How does that information fit with your results?

5) Could give some water budget, i.e. the amount of sediment needed to generate a sufficiently large anomaly. I also wonder, where the water which is exsolved actually comes from? If I understood correctly, the sediment melt can host a lot more water than the phlogopite and thus can release sufficient water on crystallization. However, is there enough water there to actually accumulate large fractions on volatile components in the melt? I am obviously not a petrologist, but many readers will also not have detailed knowledge, so a bit more detailed explanation of the sentence in lines 61/62 might help.

Reviewer #2 (Remarks to the Author):

The transport of fluids into subduction zones and the subsequent melting to produce arc volcanics is one of the key tectonic processes occurring on earth. The fluids released as the slab descends not only trigger melting, but are also thought to play a role in seismicity. Numerous minerals are thought to contribute to the fluid flux into the slab, with serpentinite often regarded as the key mineral that fosters deep melting that feeds arc volcanoes. What is less clear are the processes of fluid-triggered melting at the slab surface and overlying mantle wedge. This is a complex, non-linear process, as the melting temperatures of the materials involved are moderated by the presence of water and yet, upon melting, the water is preferentially partitioned into the melt. The presence of many phases some of which may melt, further complicates the picture. Some have argued for a straightforward release of water that lowers the solidus of the mantle wedge triggering melting. Others have proposed more complex models of upwelling mélanges of slab material, including sediments, into the overlying wedge, producing complex geochemical signatures. This paper presents a model that lies somewhere in between these two by considering the potential contribution of slab-top sediments in the melting process. However, if I have understood the paper correctly, the authors argue that the sediment melting component is not transported to the arc itself, but is considered as a fore-arc product. I am not a petrologist, but I believe the argument of the mélange melting camp is that some inputs from sediments are needed to explain the complexity of arc volcanoes, so perhaps that is a consideration for this new model?

Images of fluid release and melting beneath arcs come from seismic and Magnetotelluric (MT) soundings. This paper focusses on two MT models, one from Cascadia and one from Kyushu.

The authors have done melting experiments of representative materials (peridotite for the slab and sediment) and demonstrate that at temperatures expected within the subduction system the sediments melt producing a range of minerals (phlogopite amongst them). They use existing laboratory data of conductivity to calculate expected in-situ conductivities for Cascadia and Kyushu and create synthetic electric models from which synthetic MT responses are calculated. These responses are, in turn, inverted to generate smooth models consistent with these responses and,

finally, the models are compared to the published models derived from observed field data. This last step of inversion is important because of the inherent non-uniqueness in MT modeling: a direct comparison of the raw electrical model and the published models is not at all convincing that they are similar, but put through the filter of inversion that accounts for data resolution and model smoothing, there is reasonable agreement, at least for Cascadia.

Focusing on Cascadia, the sediment melting model splits out the source of melting into two components, a deeper primary arc melting component that, at least at the slab interface, is largely in agreement with the original interpretation of the McGary et al. MT model, and a slightly shallower component resulting from sediment melting and resulting fluids at top of the slab conductor. In the original McGary et al. paper, the large slab-top conductor was simply identified as a single feature resulting from melting triggered by serpentinite breakdown: that interpretation was the result of limited resolution and with no good reason to suggest multiple sources feeding into a single model feature. The authors argue that, in addition to the fluids and melt produced (both of which are electrically conductive), conductivity is further enhanced by the production of phlogopite, which in some studies has been argued to be electrically conductive.

Although the features A and A' are in good agreement with the McGary et al model, conductor C in Figure 3 is more prominent in this new model than the corresponding feature in the McGary MT model, and is also somewhat deeper. In the synthetic model, this feature is argued to be caused by saline fluids, whereas in the McGary paper it was viewed (and appeared to be imaged) as the upward continuation of melt transport to the arc. In this new model, melt transport is argued to be directly beneath the arc, but does not appear as a resolved feature in the model. Without exploring the model space constrained by the original data, it is hard to decide whether or not this discrepancy is significant. There is a bigger spatial disconnect between conductor C and the arc in this new model than in the prior McGary model and it is fair to point out that seismic imaging of the region (Flinders and Shen, 2017) is more consistent with the McGary model, with regions of low Vs seen in the lower crust offset to the west of Mt Ranier. Thus, the new model appears to be introducing a level of complexity that isn't required to explain the existing observations, although may well make perfect sense from a petrologic viewpoint.

Note that the authors comment that some conductors emanating from the slab have been interpreted to correlate to the basalt-eclogite transition. In the case of Cascadia, that feature is much shallower, at a depth of ~40km, and is corroborated by a seismic velocity change consistent with that transition. There is markedly less agreement between the model for Kyushu and the observations of Hata et al. The explanations given for the discrepancy are not particularly convincing. At the end of the day, the model of Hata is grounded in field observations and it is incumbent on the authors to have their model fit those observations (including the deep structure), or else argue that their model specifically does not work for this setting. It is true that Hata et al. (for technical reasons) did not impose a slab structure in their inversions. Others have shown that process to be important when inverting MT data in subduction zones. My suspicion is that inclusion of the slab would change the geometry of the conductor above the slab somewhat, but not substantially. In the Hata model the conductor appears beneath the arc, not shifted to the forearc. I don't think that the deeper structure appearing in this new model can be simply dismissed: it points to something fundamentally out of alignment with the conditions in the subduction system, in thermal structure and with the geochemistry.

The authors cite the Worzewski paper from Costa Rica which was perhaps the first to point out the lower crust/upper mantle conductor that is shifted arc-wards and which was, presumably, as motivation behind this paper. I'm puzzled why the authors didn't chose to model Costa Rica as a case study?

An issue that I don't think this is a serious concern, but deserves mentioning, is that both Cascadia and Kyushu have significant along strike variability in conductors atop the slab. The authors talk about this to some extent for Kyushu. Pommier and Evans talk about how the subduction of "anomalous" seafloor (heavily faulted or fractured, seamount of ridge structures etc) might introduce more fluids into the system through more extensive serpentinisation. Since serpentinisation is part of the proposed model, I think that fits, but I'm wondering about the effects of different sediment subduction along-strike. I'm not sure that there exists a good set of MT models in a system with marked differences in sediment transport into the subduction system, but the authors might look for a sediment starved system to examine?

A summary recommendation is that I think the authors are onto an interesting and important angle on subduction melting that may improve, or at least inform, our interpretations of process in these settings. I would like to see a more convincing explanation for the structure beneath Kyushu before the paper is accepted.

Reviewer #3 (Remarks to the Author):

The paper's claims relate to the magnetotelluric modelling of conductivity data in two subduction zones, discussed in relation to the results of petrological experiments. I am only qualified to consider the petrological aspect of the work, and this is not the focus of the paper's novel contribution. Rather, I understand that the petrological experiments discussed in the paper (which are of considerable interest and importance) are primarily presented in Foester et al (2019) a+b, in *J Asian Earth Sci and Sci. Adv.*, cited as refs 11 and 12 in this paper.

Consequently, I have not written a full report. However, I confirm that the paper's petrological dimension appears accurate and informative. The elucidation of subduction zone processes is a very complex problem of long standing, and to make progress we will certainly need to combine knowledge drawn from experimental petrology with a variety of remote sensing methods. This paper fulfils that role, and makes a valuable contribution to the field, facilitated by the recent experimental work cited above.

REVIEWER COMMENTS

Reviewer #1 (Remarks to the Author):

The paper ‘Melting of subducted sediments reconciles geophysical image of subduction zones’ presents a study which aims to explain a globally observed conductivity anomaly in the trench ward side of the volcanic arc. The authors explain the conductivity anomaly to be caused by exsolution of saline fluids during crystallization of a phlogopite-pyroxenite metasome, which in turn is related to sediment melting. The authors present a synthetic MT modelling study, in which they generate a conductivity model based on petrophysical assumptions including the phlogopite-pyroxenites metasome and temperature models of the Cascadia and Kyushu subduction zone. For these conductivity models, MT forward responses are calculated and then inverted. The resulting conductivity model is then compared to conductivity models derived from real data for these two subduction zones. Agreement of the conductivity anomaly observed in the real and synthetic inversion model is then interpreted as evidence that the conductivity anomaly is related to sediment melting. Sediment melting is also proposed as a possibility to explanation for the existence of K-rich volcanic rocks in conjunction with slab-rollback events.

The paper is certainly of sufficient interest and novel to be published by a high impact journal like Nature Geoscience since it explains a globally observed anomaly of enigmatic origin and contributes to further understand, define and quantify hydration and dehydration processes of subduction zone. It is also clearly structured and well written and documented, in fact I enjoyed reading the paper. I therefore recommend it for publication after some amendments or discussions listed below are included in the document.

1) I think the paper would benefit from reasoning the strategy that the author’s follow in presenting their ‘evidence’. The authors go from a temperature and compositional model (geological model) to a resistivity model (physical parameter model), then translate that into data (data space) and reinvert the data to a physical parameter model and compare this with the derived observed physical parameter model. I do appreciate that the authors take into consideration that the shape of the physical parameter model very much depends on the data and inversion algorithm as it gives us only a biased or ‘smeared’ look at the Earth and it is that image which we should compare to the inversion model of the data. However, I think the paper would benefit if you would explain your overall strategy and most importantly reason why you do this. I think this explanation is necessary since you deviate from the trodden path in which one takes the physical parameter inversion model and interprets the anomalies with regard to temperature and/or composition and/or fluid content.

This is a helpful comment - we think the difference between the forward modelled resistivities and the subsequent inversion of those data is very important so we are very happy to clarify this. We note that this comment also relates to Reviewer 2’s Point 2 below, where Reviewer 2 comments that the final step of inverting the forward modelled data is important because of the inherent non-uniqueness of MT modelling. Indeed, this is the reason for going through this process. The differences between the forward models shown in SI Fig. 3 and the subsequent inversions shown in Figures 3 and 4 of the manuscript illustrate how important it is to consider not only a theoretical resistivity distribution but also how this resistivity distribution would be modelled with real MT data.

The final two sentences of Paragraph 3 in Section 2 have been re-written as:
‘Theoretical resistivity structures were calculated based upon simplified subduction zone compositions of mantle peridotite, phlogopite-pyroxenite metasome, and 1% sediment melt, 1% arc melt, and 1% saline fluid in the appropriate regions (see Methods, Supplementary Table 1 and Figs

2-4). The MT responses of the resistivity structures were forward modelled and these MT responses were then inverted. This procedure was chosen because MT inversions are non-unique and it allowed us to directly compare our inversion of synthetic data with the inversion of field data.'

2) Please discuss the uncertainties in resistivity model that is generated (supplementary section). You make various assumptions about the composition, sediment volume and temperature (the effect of melt volume has been investigated). A large uncertainty will arise from the effective medium theory chosen to calculate the overall bulk conductivity of the mix. I would like to see some more detailed explanation/reasoning, how the resistivity model may vary depending on different assumptions/choices made and why the authors have chosen the particular values of the parameters. Also, do you see any possibility to derive an exsolved water volume from the observations directly which could be applied to MT studies on other subduction zones, i.e. convert the physical parameter to relevant geological parameters and if so, with which trade offs? This would be particularly interesting as it would help to interpret MT resistivity models at other subduction zones.

The final section in the SI has been expanded to discuss the uncertainties in the synthetic resistivity model in more detail. As noted by the reviewer, there are uncertainties in converting compositions and temperatures to predicted resistivities, largely due to experimental uncertainties and incomplete knowledge of subsurface composition, temperature and rock geometry. Unfortunately, most of these uncertainties are inherent to the calculations due to our current state of knowledge but we agree with the reviewer that they should be described in more detail. The SI text has been changed (reproduced below) and an additional figure (SI Figure 5) has been added to show the impact of the choice of geometry of the conductive melt/fluid phase.

Given the interplay between composition, temperature, geometry and bulk conductivity, and the uncertainties in all these parameters, unfortunately I do not think it is currently likely that the volume of exsolved water could be interpreted directly from the MT data. In particular, once fluid is interconnected, an increase in the volume of that fluid has little additional effect on conductivity. However, the analysis that we have applied here could certainly be applied to other subduction zones to determine whether an interconnected fluid phase is consistent with the MT data. Hopefully, as experimental data and constraints on compositions and temperatures improve, our ability to make quantitative interpretations of geophysical models will also improve.

Additional text in SI:

'Compositions and temperatures in the subduction system are not accurately known. In addition, the conversion of these compositions and temperatures to electrical resistivities requires choosing experimental conductivity models (Table 1) and making assumptions about rock geometry, introducing additional uncertainty into the analysis. In our analysis we make conservative choices for these parameters where possible, using a very simplified compositional model and reliable thermal models. In all regions where fluid or melt is present, modelling was carried out assuming 1% fluid/melt but tests for the Cascadia system were also run with 0.5% and 5% melt/fluid. As shown in Figure 4, these different melt proportions altered the magnitude of the conductors but did not strongly affect the overall conductivity structure.

Given that we are modelling a system with strong conductivity contrasts between the melt/fluid phase and the solid phase, the geometric model that is used to calculate total resistivity is an important parameter. The influence of the conductive fluid/melt phase on bulk conductivity will be controlled by its interconnectivity. Figure 5 illustrates this point by showing the large contrasts in conductivities between the Hashin-Shtrikman (HS) upper and lower bounds for the peridotite plus melt, peridotite plus fluid, and phlogopite-pyroxenite plus melt compositions. The HS upper bound calculates bulk conductivity assuming that the most conductive phase (here the fluid/melt) is perfectly connected, while the lower bound assumes it is disconnected. Experimental and theoretical

models suggest that 1% fluid/melt should be well connected (Holtzman 2016; Laumonier et al., 2017), showing that the actual conductivity is likely to approximate the HS upper bound. Given that the fluid/melt interconnectivity can be estimated, we used a modified Archie's Law (Glover, 2010) to calculate the bulk rock conductivity. We assumed good interconnectivity of the fluid/melt phase, as shown by the close approximation of the Archie's Law conductivity to the HS upper bound. The Cascadia model test run with 0.5% melt/fluid shown in Figure 4(a) can also be considered to approximate the conductivity response of a system containing 1% melt/fluid with poorer interconnectivity, showing that the main features are reproduced.'

The Methods section was also revised to say, 'A modified Archie's Law was used to geometrically combine the components⁴⁰, assuming good connectivity of the conductive phase as expected for 1% melt or fluid (Laumonier et al.,2017; Supplementary Fig. 5).'

Supplementary Figure 5 is as below:

Fig. 5: Comparison between bulk conductivities calculated using Hashin-Shtrikman (HS) lower and upper bounds and Archie's Law. As the 1% melt/fluid was assumed to be well connected, the bulk conductivity calculated with Archie's Law approximates the HS upper bound.

3) I am somewhat puzzled by the inversion results of the synthetic Cascadia data. There is a strong anomaly in the predicted conductivity structure associated with the phlogopite/pyroxenite sediment melt. However, in the inversion model the high conductivity anomaly is not visible at the same location anymore but seems to have moved towards profile kilometer 10 or 20 and to a depth of 90 km. This should be explained, particularly since I do not see the same effect for the Kyushu case. Also, it would be great if you could include a bit more detail about the number of Mt stations you simulate as well as the frequency range in the data. What was the starting model that you chose?

Information about the period range (17 periods over the range $10-10^5$ s) and starting model (100 ohm m half-space) has been added to the final paragraph of the Methods section. Both Cascadia and Kyushu models were run with 25 synthetic stations spaced at 4 km intervals.

I think the reviewer here is referring to the strong conductor labelled A on Figure 3. This is an interesting point and we agree this conductor is much stronger in the inverse model than in the forward (predicted) model. We believe that this reflects resolution limitations in MT. A comparison between the predicted resistivity structure in SI Fig. 3(a) and the inverted resistivity structure in Figure 3 shows that the resistivity in the inverted model is lower at depths greater than ~75 km but higher at shallower depths. This is illustrated in Figure R1.3.1 below which shows resistivity-depth profiles from the forward model (blue) and inverted model (red) from 0 to 90 km depth at the centre point of the model (0 km). The best resolved feature in MT is the conductance (conductivity x

depth). The conductance of the forward (predicted) profile shown below is $8.7e3$ while that of the inverse profile is $1.8e4$, confirming that the MT responses of these two resistivity structures are very similar. This is further confirmed by the data fits for the inverse model, which are also shown below (Figure R1.3.2). I do think this feature is also in the Kyushu model (consider Reviewer 2's comment 5), but the conductor is less strong because the modelled conductance is able to extend to greater depth since the resistive slab has not been imposed.

Fig. R1.3.1: Resistivity-depth profiles at the centre point of the Cascadia forward (predicted) model (blue line) and inverse model (red line). The total conductance through this section has been maintained but the low resistivities have been pushed to a greater depth in the inverse model.

Fig. R.1.3.2: Model fits for the inverse Cascadia model (circles are the forward modelled data and lines are the inverse model responses). Data fits are good at all stations and at all periods, showing that the difference between the forward and inverse model is due to inherent uncertainties in the MT method.

Additional detail about this has been added to the end of Paragraph 3 of Section two as follows:

‘Non-uniqueness in the inversion process has converted the broad, moderately low resistivity zone caused by arc melts in the forward model (Supplementary Fig. 3(a)) to a thinner, lower resistivity zone at greater depth in the inverse model (Region A in Fig. 3), while maintaining a good fit to the data.’

4) In the paper of Worzewski et al., the anomaly was always found at a consistent depth and distance from the volcanic arc, independent of the convergence rate, subducting plate age or style (erosional/accretionary) of the subduction. How does that information fit with your results?

Actually, we believe that an examination of Figure 4 in Worzewski et al. 2010 shows the location of the anomaly does vary. In that compilation it lies between 20-80 km in front of the arc and at 20-50 km depth. Our results for Cascadia and Kyushu are located within these observed ranges. As we show in Fig. 1, the metasome will form when the slab-surface temperature crosses the sediment solidus. Since not all slab surface temperatures reach the solidus (range between green line and sediment melting solidus in Fig. 1) the cold subduction zones probably do not have fore-arc metasomes.

5) Could give some water budget, i.e. the amount of sediment needed to generate a sufficiently large anomaly. I also wonder, where the water which is exsolved actually comes from? If I understood correctly, the sediment melt can host a lot more water than the phlogopite and thus can release sufficient water on crystallization. However, is there enough water there to actually accumulate large fractions on volatile components in the melt? I am obviously not a petrologist, but many readers will also not have detailed knowledge, so a bit more detailed explanation of the sentence in lines 61/62 might help.

We thank the reviewer to raise the important question how volatile contents of high-pressure experiments compare to natural subduction zones. In subduction zones, the sediment layer largely de-waters before it reaches depths of partial melting. However, located beneath sediment layer, the oceanic lithosphere is serpentinized and also colder than the sediment cover above (which is closer to the mantle wedge, and thus, heats first). Hence, the interior of the subducted slab heats later and releases volatiles that then rise through the sediment layer. We added references and expanded this part:

“The high-pressure experiments do not account for fluid loss before sediment melting and volatile contents could be overestimated. Nevertheless, in subduction zones, the volatile content of this sediment melt should be above >5% due to the continuous devolatilization of serpentinites^{4,13} that underly the sediments, which are an additional fluid source that is not accounted for in the reaction experiments. Since fluids have high solubility in silicic melt¹⁴ and partial melting of sediments produces <10% melt¹¹, even low amounts of fluid (~0.5-1%) from the underlying slab translate to 5-10% volatiles within the sediment melt.”

Reviewer #2 (Remarks to the Author):

1. The transport of fluids into subduction zones and the subsequent melting to produce arc volcanics is one of the key tectonic processes occurring on earth. The fluids released as the slab descends not

only trigger melting, but are also thought to play a role in seismicity. Numerous minerals are thought to contribute to the fluid flux into the slab, with serpentinite often regarded as the key mineral that fosters deep melting that feeds arc volcanoes. What is less clear are the processes of fluid-triggered melting at the slab surface and overlying mantle wedge. This is a complex, non-linear process, as the melting temperatures of the materials involved are moderated by the presence of water and yet, upon melting, the water is preferentially partitioned into the melt. The presence of many phases some of which may melt, further complicates the picture. Some have argued for a straightforward release of water that lowers the solidus of the mantle wedge triggering melting. Others have proposed more complex models of upwelling mélanges of slab material, including sediments, into the overlying wedge, producing complex geochemical signatures. This paper presents a model that lies somewhere in between these two by considering the potential contribution of slab-top sediments in the melting process. However, if I have understood the paper correctly, the authors argue that the sediment melting component is not transported to the arc itself, but is considered as a fore-arc product. I am not a petrologist, but I believe the argument of the mélange melting camp is that some inputs from sediments are needed to explain the complexity of arc volcanoes, so perhaps that is a consideration for this new model?

We absolutely agree with the reviewer that the subduction process is complex and the compositions in different parts of the subduction system are beyond what we are able to include in our model. There is no doubt that some melted subducted sediment reaches the arc in many subduction zones and we don't mean to suggest otherwise. Within our model, the volume of sediment melt that reaches the arc is likely to depend on the subduction zone geotherm and geometry, as illustrated by the section discussing K-rich volcanism and the sentences at the end of the Reaction Experiments section which state '...in steeply-dipping subduction zones the sediment melting and arc melting regions may essentially overlie one another. In such a situation the metasome is unlikely to develop at depth but arc magmas may be more K-rich as they have a higher component of sediment melt which would have been otherwise trapped within the fore-arc metasome.' The involvement of sediment in arc melts is also mentioned in the abstract.

We have altered the final sentences of Paragraph 1 of Section 1 to make this more clear, with this section now stating, 'The remaining portion of the subducted sediments will melt, often making a chemically important contribution to arc melts². As discussed below, in many subduction zones this will occur at depths between fore-arc serpentinites and the source region of arc magmas.'

Images of fluid release and melting beneath arcs come from seismic and Magnetotelluric (MT) soundings. This paper focusses on two MT models, one from Cascadia and one from Kyushu.

2. The authors have done melting experiments of representative materials (peridotite for the slab and sediment) and demonstrate that at temperatures expected within the subduction system the sediments melt producing a range of minerals (phlogopite amongst them). They use existing laboratory data of conductivity to calculate expected in-situ conductivities for Cascadia and Kyushu and create synthetic electric models from which synthetic MT responses are calculated. These responses are, in turn, inverted to generate smooth models consistent with these responses and, finally, the models are compared to the published models derived from observed field data. This last step of inversion is important because of the inherent non-uniqueness in MT modeling: a direct comparison of the raw electrical model and the published models is not at all convincing that they are similar, but put through the filter of inversion that accounts for data resolution and model smoothing, there is reasonable agreement, at least for Cascadia.

Thanks for this comment, and we agree that considering the inverse MT response is a really important step when comparing theoretical electrical structures and inverted models of real data. We note that this comment is also relevant to Point 1 of Reviewer 1.

Focusing on Cascadia, the sediment melting model splits out the source of melting into two components, a deeper primary arc melting component that, at least at the slab interface, is largely in agreement with the original interpretation of the McGary et al. MT model, and a slightly shallower component resulting from sediment melting and resulting fluids at top of the slab conductor. In the original McGary et al. paper, the large slab-top conductor was simply identified as a single feature resulting from melting triggered by serpentinite breakdown: that interpretation was the result of limited resolution and with no good reason to suggest multiple sources feeding into a single model feature. The authors argue that, in addition to the fluids and melt produced (both of which are electrically conductive), conductivity is further enhanced by the production of phlogopite, which in some studies has been argued to be electrically conductive.

3. Although the features A and A' are in good agreement with the McGary et al model, conductor C in Figure 3 is more prominent in this new model than the corresponding feature in the McGary MT model, and is also somewhat deeper. In the synthetic model, this feature is argued to be caused by saline fluids, whereas in the McGary paper it was viewed (and appeared to be imaged) as the upward continuation of melt transport to the arc. In this new model, melt transport is argued to be directly beneath the arc, but does not appear as a resolved feature in the model. Without exploring the model space constrained by the original data, it is hard to decide whether or not this discrepancy is significant. There is a bigger spatial disconnect between conductor C and the arc in this new model than in the prior McGary model and it is fair to point out that seismic imaging of the region (Flinders and Shen, 2017) is more consistent with the McGary model, with regions of low Vs seen in the lower crust offset to the west of Mt Ranier. Thus, the new model appears to be introducing a level of complexity that isn't required to explain the existing observations, although may well make perfect sense from a petrologic viewpoint.

We agree with the reviewer that our model is a very simplified view of the full compositional complexities likely to exist in the subduction system. We have added a sentence to the end of Paragraph 3 of Section 2 stating 'These models are simplified and do not seek to model the full complexities of potential subduction zone conductive anomalies but are specifically designed to model the anomalies arising from the melting of subducted sediments.' The depth of conductor C is due to our simplified assumption that the saline fluids do not ascend beyond 10 km depth. If pathways exist allowing some fluid to ascend to higher depths or to migrate laterally, the shape and extent of the conductor will differ and may resemble the McGary et al. (2014) model more closely. We did not run such tests as they would have been getting into the realm of pattern matching rather than examining the hypothesis, but we did amend the end of the first paragraph of the Magnetotelluric Modelling section of the Methods to say, 'Throughout the model region, melt and fluid phases extended to a depth of 10 km, assumed to be the brittle-ductile transition, though this may vary by location and upper crustal structures may allow for additional upward and lateral fluid flow that would change the shallow resistivity structures.'

An important point is that arc melt is not strongly conductive at upper crustal temperatures (Supplementary Fig. 2), so it is hard to interpret the very low resistivities modelled by McGary et al. (2014) in the upper crust as arc melt at reasonable melt fractions. Indeed, we included arc melt in our synthetic models to 10 km depth beneath the arc volcano but the conductor associated with the saline fluid dominates the models at these depths because the saline fluid is so much more conductive than the arc melt. This is mentioned in Paragraph 4 of Section 2 (originally Paragraph 3) where we state, 'Although arc melt is included in a column extending up to the arc volcano, this region is not associated with a strongly conductive anomaly. This is primarily because the conductivity of arc melt is significantly lower than saline fluid at crustal and uppermost mantle temperatures (Supplementary Fig. 2).' To clarify further, we have added a sentence to Paragraph 3

of Section 2, stating, 'In the temperature range of interest, the phlogopite-pyroxenite is more conductive than peridotite and the saline fluid is more conductive than arc melt.'

We think that Flinders and Shen (2017) seismic model actually illustrates this point quite nicely. Our model suggests that there should be arc melt beneath Mt Rainier and saline fluid to the west. In seismic models both arc melt and saline fluids will produce slow velocity anomalies. In MT models the saline fluid will produce a much stronger conductivity anomaly. Therefore we would expect a slow seismic anomaly but no strong conductive anomaly directly beneath Mt Rainier and a slow seismic anomaly and a strong conductive anomaly to the west. The Flinders and Shen (2017) seismic model shows slow anomalies both beneath and 15-20 km to the west of Mt Rainier, supporting our model. The slow seismic anomaly directly beneath Mt Rainier is not associated with a conductive anomaly, while the western slow seismic anomaly is approximately coincident with the strong conductive anomaly of McGary et al. (2014), also supporting our model.

We have included some discussion of this seismic model in the text, adding the following to Paragraph 4 of Section 2: 'Seismic models of the region (Flinders and Shen, 2017) support this interpretation, with slow seismic velocities modelled both directly beneath and ~15-20 km trenchward of Mt. Rainier, aligned with our proposed regions of arc melt and saline fluids. Since saline fluids are more conductive than arc melts at crustal temperatures, the conductive response of the saline fluids dominate the MT model.'

4. Note that the authors comment that some conductors emanating from the slab have been interpreted to correlate to the basalt-eclogite transition. In the case of Cascadia, that feature is much shallower, at a depth of ~40km, and is corroborated by a seismic velocity change consistent with that transition.

We absolutely agree that there will be other features in subduction zones, including the basalt-eclogite transition, that will create conductivity anomalies that we are not investigating here. We have re-worded the start of Section 2 to make it clear that the alternative interpretations listed (like the basalt-eclogite transition) are not solely related to the fore-arc conductor we are investigating. We have also added a sentence to the end of the second paragraph of Section 2 stating:

'These models are highly simplified and do not seek to model the full complexities of potential subduction zone conductive anomalies but are specifically designed to model the anomalies arising from the melting of subducted sediments.'

5. There is markedly less agreement between the model for Kyushu and the observations of Hata et al. The explanations given for the discrepancy are not particularly convincing. At the end of the day, the model of Hata is grounded in field observations and it is incumbent on the authors to have their model fit those observations (including the deep structure), or else argue that their model specifically does not work for this setting. It is true that Hata et al. (for technical reasons) did not impose a slab structure in their inversions. Others have shown that process to be important when inverting MT data in subduction zones. My suspicion is that inclusion of the slab would change the geometry of the conductor above the slab somewhat, but not substantially. In the Hata model the conductor appears beneath the arc, not shifted to the forearc. I don't think that the deeper structure appearing in this new model can be simply dismissed: it points to something fundamentally out of alignment with the conditions in the subduction system, in thermal structure and with the geochemistry.

Thanks for this comment. In response we have looked at these discrepancies in more detail. We believe they are largely due to a lack of resolution in the MT data of Hata et al. (2017) at those depths, likely compounded by the different MT inversion schemes used. Below we provide a

detailed response to the reviewer, followed by a description of the changes we have made to the manuscript to explain this.

The data in the resistivity model of Hata et al. (2017) come from two different sources. The long-period MT data are described in Hata et al. (2012) and Hata et al. (2015) and are ‘network-MT’ data. The network-MT data utilise commercial telephone wires for electric field measurements with dipole lengths of ~10-30 km, producing MT data from periods ~500-40,000 s. There are five of these network-MT stations on the Kirishima profile with periods ranging from ~500-20,000 s (Hata et al., 2012, Figure 6).

Hata et al. (2017) add geomagnetic depth sounding (GDS) data to the network-MT data. The GDS stations are more densely spaced (as little as 10 km spacing) but only extend to shorter periods (4.6-1000 s) so they have poorer depth resolution than the network-MT data. Due to the nature of GDS data, they are not sensitive to vertical conductivity gradients. Given that the main area of disagreement between our synthetic model and the model of Hata et al. (2017) is an approximately flat-lying conductor at depths greater than ~60 km, the GDS data are unlikely to provide much additional resolution of this feature. Hata et al. (2017) used the 3D network-MT model of Hata et al. (2015) as their starting model and inverted only the new GDS data, they did not run a combined network-MT/GDS inversion. The features below ~50 km depth in Hata et al. (2017) are essentially unchanged from those in Hata et al. (2015) (Fig. R2.5.1).

Fig R2.5.1: Final inversion from Hata et al. (2015) (Figure 7) from the long-period network-MT data only (left) and the final inversion from Hata et al. (2017) (Figure 3) from GDS inversion from a long-period network-MT starting model (right). The shorter period GDS data have mainly resolved features at <50 km depth, while deeper features are mostly unchanged.

The disagreements between the Hata et al. (2017) model and our synthetic model are mostly at depths >60 km, where resolution is essentially from the longer-period network-MT data. Several resolution tests shown in the various papers of Hata et al. show that the network-MT model is poorly resolved at these depths. Hata et al. (2015) Figure 3 (Fig. R2.5.2 below) shows checkerboard tests run on the MT model. A conductor ($\log \rho = 0.5$ ohm m) was emplaced from 60-100 km depth across the Kirishima region. The resulting inversion did not reproduce the conductor but instead modelled the region with much higher resistivity ($\log \rho > 1.5$ ohm m) while producing a low rms error of 0.96, indicating that the model has poor resolution to a conductor at this depth. The authors further tested resolution by replacing the conductor beneath Kirishima in their final model with a more resistive region from 45 km depth (Fig. R2.5.3). The data are more closely fit by a

conductor than a resistor but the overall change in rms error is small (0.1). These results and the data fits shown in Hata et al. (2015) Figure 6 (Fig. R2.5.4) and Hata et al. (2012) Fig. 4 also show that there are substantial misfits between the observed data at periods $>10,000$ s and their preferred model. Therefore, it is reasonable to conclude that the model is poorly resolved at depths greater than ~ 50 km.

Fig. R2.5.2: Figure 3 from Hata et al. (2015). Checkerboard tests for resolution of the 3D network-MT model. (f) shows that a strong conductor extending from 60-100 km depth beneath Kirishima is not reproduced in the inversion, showing that resolution is poor in this model region.

Fig. R2.5.3: Excerpts from Figure 9 of Hata et al. (2015). The authors ran a forward resolution test by increasing the resistivity inside the blue box from 45 to 150 km depth. The plots on the right show the impedance tensor elements for the observed data (red), preferred inverse model (blue) and forward test (green). The impact of changing resistivity from 45 km depth is minor and mostly at periods $>10,000$ s. The more conductive preferred model provides a better data fit than the more resistive forward model. Importantly, the preferred inverse model provides a poor fit to the observed Ty data at periods $>10,000$ s and the data suggest that the true earth is more conductive than the inverse model.

Fig. R2.5.4: Excerpt from Figure 6 of Hata et al. (2015), showing the phase residual (observed minus model). In the Kirishima region, these phase misfits are ~5-10 degrees. Higher phases in the data than the model suggest that the Earth conductivity increases with depth more than the model.

The authors of Hata et al. (2015) show some additional model tests in Appendix 1 with varying constraints on slab architecture and resistivity (Fig. R2.5.5). Results show that the depth extent of the conductor varies considerably with these different model constraints while the misfit remains low, further demonstrating that resolution of the model in this region is poor.

Fig. R2.5.5: Figure A1 from Hata et al. (2015). Model test of the slab resistivity illustrate that model resolution at depths >60 km is poor and that the data are compatible with much lower resistivities at these depths than in the preferred model.

Our synthetic models were run with a much denser data coverage than exists in the real data from Kyushu, with full MT data at every station, stations spaced every 4 km, and periods at all stations extending from 10 to 100,000 s. Therefore, the main reason for the differences between the models is likely to be the difference in model resolution at depths >60 km. However, even with the better data coverage, the results of the synthetic inversion also suggest that the model structure leads to some inherent uncertainty in resistivity at these depths. Figure R2.5.6 shows the difference between the forward models ((a) and (c)) and the resulting inverse models ((b) and (d)), with and without arc melt included in the forward models, respectively. In both cases, resistivities in the inverse models are significantly lower at depths greater than ~100 km than in the forward models.

Therefore, we conclude that differences between the Hata et al. (2017) model and our synthetic model at depths >50 km are largely due to poor model resolution. The different regularization approaches in the different inversion schemes appear to deal with this poor resolution differently:

Siripunvaraporn et al. (2004) has imposed a resistor whereas MARE2DEM has imposed a conductor. I am not aware of any studies that have directly compared these inversion codes but model tests of Siripunvaraporn et al. (2005) in Miensopust et al. (2013, GJI) suggest that this code tends to place resistors in regions of poor resolution.

Figure R2.5.6: Forward and inverse synthetic models of the Kyushu system containing arc melt (a and b) and without arc melt (c and d). In both cases, the resistivity modelled in the inverse models at depths greater than ~ 100 km is lower than in the forward model, highlighting poor resolution at these depths.

In the manuscript, we have removed the previous text explaining the differences between our model and the published Hata et al. (2017) model and replaced it with the following:

‘At greater depths the agreement between the synthetic and original Kyushu models is poorer and the synthetic model shows a strong conductor from ~ 60 km depth that is not observed in the original model. This discrepancy appears to be mainly due to a lack of model resolution at this depth. The original Kyushu model (Fig. 4(a)¹⁹) is derived from widely-spaced network-MT stations that extend to periods of $>20,000$ s (Hata et al., 2015) together with geodynamic depth sounding data that are denser but have a more limited period range (1000 s). Resolution of features at depths >50 km is largely from the network-MT data and tests run by Hata et al., (2015) show that resistivity is poorly resolved at these depths and resistivities up to an order of magnitude higher and lower than those in the preferred model produce acceptable data fits. The synthetic model contains a broader period range (10 - 10^5 s) at more densely spaced stations (4 km) but still shows poor resolution in this part of the model, highlighted by the difference between the forward (Supplementary Fig. 3) and inverse (Fig. 3) models. The original and synthetic models were run with different inversion schemes that appear to have responded to this lack of resolution by imposing more resistive and more conductive features, respectively.’

We have also extended the shaded region on Figure 4 to cover depths greater than 50 km and have changed the end of the figure caption to read, 'The synthetic model approximately reproduces the conductor extending to ~530 km depth just trenchward of Kirishima volcano (red triangle). Shading at depths >50 km reflects poorer model resolution at these depths.'

6. The authors cite the Worzewski paper from Costa Rica which was perhaps the first to point out the lower crust/upper mantle conductor that is shifted arc-wards and which was, presumably, as motivation behind this paper. I'm puzzled why the authors didn't chose to model Costa Rica as a case study?

A good question! The main answer is that the motivation for this work actually came from thinking about the possible links between the petrologic experiments and the somewhat iconic McGary et al. (2014) Cascadia model. The Worzewski et al. (2011) paper came in to our thinking later as we considered the broader implications. As mentioned in the manuscript (Section 2) Cascadia and Kyushu were chosen because they have reliable geotherm models, reliable MT models and, in the case of Kyushu, because of the along-strike variability in K-enrichment. There is no reason why Costa Rica could not be examined in the future.

7. An issue that I don't think this is a serious concern, but deserves mentioning, is that both Cascadia and Kyushu have significant along strike variability in conductors atop the slab. The authors talk about this to some extent for Kyushu. Pommier and Evans talk about how the subduction of "anomalous" seafloor (heavily faulted or fractured, seamount of ridge structures etc) might introduce more fluids into the system through more extensive serpentinisation. Since serpentinisation is part of the proposed model, I think that fits, but I'm wondering about the effects of different sediment subduction along-strike. I'm not sure that there exists a good set of MT models in a system with marked differences in sediment transport into the subduction system, but the authors might looks for a sediment starved system to examine?

We agree and we did consider this but also could not find a good set of MT models to investigate the question. According to our model, the size of the metasome should depend on the amount of sediment subducted, so this should be a testable hypothesis. Hopefully, if our model can be published, this will be an idea that people will consider when interpreting data in the future. We have added a comment stating that MT models of sediment-starved subduction systems would be a good test of our model to the end of Paragraph 1 of Section 3.

A summary recommendation is that I think the authors are onto an interesting and important angle on subduction melting that may improve, or at least inform, our interpretations of process in these settings. I would like to see a more convincing explanation for the structure beneath Kyushu before the paper is accepted.

Reviewer #3 (Remarks to the Author):

The paper's claims relate to the magnetotelluric modelling of conductivity data in two subduction zones, discussed in relation to the results of petrological experiments. I am only qualified to consider the petrological aspect of the work, and this is not the focus of the paper's novel contribution. Rather, I understand that the petrological experiments discussed in the paper (which are of considerable interest and importance) are primarily presented in Foester et al (2019) a+b, in J Asian Earth Sci and Sci. Adv., cited as refs 11 and 12 in this paper.

Consequently, I have not written a full report. However, I confirm that the paper's petrological dimension appears accurate and informative. The elucidation of subduction zone processes is a very

complex problem of long standing, and to make progress we will certainly need to combine knowledge drawn from experimental petrology with a variety of remote sensing methods. This paper fulfils that role, and makes a valuable contribution to the field, facilitated by the recent experimental work cited above.

REVIEWERS' COMMENTS

Reviewer #1 (Remarks to the Author):

Thank you for addressing all points raised in the review.

Reviewer #2 (Remarks to the Author):

In revising their manuscript the authors have taken the comments I raised seriously and have been thorough in their consideration. In particular, the authors have presented detailed work to document possible reasons for the discrepancy between their model for Kyushu and that of Hata et al. I will say that my concerns remain with the deeper parts of their Kyushu model which seem unreasonably conductive, but these are now blanked out and are not a primary part of the discussion, so I think that is ok.

On the whole, I think the idea presented is interesting and worth publishing. As with all such models, it will be open to criticism of the choices of parameters, but that is what it is. As an example, I'd point to the observations at Mount St Helens by Bedrosian et al. (Nature Geoscience, 2018) which is perhaps the best image of crustal melt distribution around an arc volcano in the literature, and which presents clear evidence for conductive melts within the crust. The comments on line 129 referring to the work of Flinders and Chen is a little misleading inasmuch as it flies in the face of the interpretation of the seismic model made by Flinders and Chen and, again, contradicts the point made above that crustal melts can be conductive. There is no argument that saline fluids in the crust will be conductive (they were the explanation given by McGary et al for their feature D), but Flinders and Chen place their model in a more regional context of melt delivery and crustal storage. I'd also point out again that feature C is quite bit deeper in the crust than the anomaly seen by McGary et al. who also point out that the thermal model which was a simple overlay on their resistivity model is a steady state solution and that a fully dynamic solution would be needed to understand the melt temperature on ascent. Finally, lab studies on melts are constantly evolving and the results of Pommier and Garnero (JGR, 2014) add Na into the mix as an element that can dramatically enhance the conductivities of incipient melts. Still, this is splitting hairs to some extent, serves to highlight the complexity of the system, and is fodder for commentary on the paper.

So, in summary, I could go back and forth with the authors on this, but should probably stop here and let the paper move forwards to publication.

Reviewer 2:

In revising their manuscript the authors have taken the comments I raised seriously and have been thorough in their consideration. In particular, the authors have presented detailed work to document possible reasons for the discrepancy between their model for Kyushu and that of Hata et al. I will say that my concerns remain with the deeper parts of their Kyushu model which seem unreasonably conductive, but these are now blanked out and are not a primary part of the discussion, so I think that is ok.

On the whole, I think the idea presented is interesting and worth publishing. As with all such models, it will be open to criticism of the choices of parameters, but that is what it is. As an example, I'd point to the observations at Mount St Helens by Bedrosian et al. (Nature Geoscience, 2018) which is perhaps the best image of crustal melt distribution around an arc volcano in the literature, and which presents clear evidence for conductive melts within the crust. The comments on line 129 referring to the work of Flinders and Chen is a little misleading inasmuch as it flies in the face of the interpretation of the seismic model made by Flinders and Chen and, again, contradicts the point made above that crustal melts can be conductive. There is no argument that saline fluids in the crust will be conductive (they were the explanation given by McGary et al for their feature D), but Flinders and Chen place their model in a more regional context of melt delivery and crustal storage. I'd also point out again that feature C is quite bit deeper in the crust than the anomaly seen by McGary et al. who also point out that the thermal model which was a simple overlay on their resistivity model is a steady state solution and that a fully dynamic solution would be needed to understand the melt temperature on ascent. Finally, lab studies on melts are constantly evolving and the results of Pommier and Garnero (JGR, 2014) add Na into the mix as an element that can dramatically enhance the conductivities of incipient melts. Still, this is splitting hairs to some extent, serves to highlight the complexity of the system, and is fodder for commentary on the paper.

Response:

We are grateful to Reviewer 2 for their continued analysis. We agree that our reference to Flinders and Shen (2017) could be misleading so we have re-written this sentence to make it clear that the seismic interpretation was different, as follows: 'Seismic models of the CAFE line region are consistent with this interpretation, with slow seismic velocities modelled both directly beneath and ~15-20 km trenchward of Mt. Rainier. These slow velocities align with our proposed regions of arc melt and saline fluids, although the interpretation of the seismic model was that they are both caused by melt.'

We also agree with the reviewer's other points about interpretation based on different compositions, dynamic thermal models, and different MT models. Through this submission, we do not seek to negate alternative interpretations, including that crustal melts can be conductive as interpreted by Bedrosian et al. (2018). Instead, we are hoping to highlight to the community the importance of considering phlogopite-pyroxenites and exsolved saline fluids as a factor in their interpretations.